# Blockchain-Based Coordination: Assessing the Expressive Power of Smart Contracts [†]

**Giovanni Ciatto [1]** , **Stefano Mariani [2],\*** , **Alfredo Maffi [3]** and **Andrea Omicini [1]**

[1]   Department of Computer Science and Engineering, University of Bologna, 47521 Cesena, Italy; giovanni.ciatto@unibo.it (G.C.); andrea.omicini@unibo.it (A.O.)

[2]   Department of Sciences and Methods of Engineering, University of Modena and Reggio Emilia, 42122 Reggio Emilia, Italy

[3]   Mark One S.R.L., 47521 Cesena, Italy; alfredo.maffi@studio.unibo.it

\*   Correspondence: stefano.mariani@unimore.it

†   This paper is an extended version of our paper published in 2nd International Workshop on Blockchain Technologies for Multi-Agent Systems (BCT4MAS), and later appeared in the Communications in Computer and Information Science book series (CCIS, volume 1047).

**Abstract:**  A common use case for blockchain smart contracts (SC) is that of governing interaction amongst mutually untrusted parties, by automatically enforcing rules for interaction. However, while many contributions in the literature assess SC *computational* expressiveness, an evaluation of their power in terms of *coordination* (i.e., governing interaction) is still missing. This is why in this paper we test mainstream SC implementations by evaluating their expressive power in coordinating both inter-users and inter-SC activities. To do so, we exploit the archetypal LINDA coordination model as a benchmark—a common practice in the field of *coordination models and languages*—by discussing to what extent mainstream blockchain technologies support its implementation. As they reveal some notable limitations (affecting, in particular, coordination between SC) we then show how Tenderfone, a custom blockchain implementation providing for a more expressive notion of SC, addresses the aforementioned limitations.

**Keywords:** blockchain; smart contracts; coordination; LINDA; Tenderfone

## 1. Introduction

The blockchain concept and technology is rapidly shaking up many different areas in both academic research and industrial practice. Its ability to secure both data and computation in a distributed setting with no need to rely on reciprocal trust or a centralised authority is an appealing way to re-think current applications and functionalities requiring a middle-man, as well as to promote brand-new business opportunities—such as, cryptocurrencies.

Over the past years the blockchain has shown to naturally fit many heterogeneous application domains, such as supply chain for tracking goods [1], healthcare for auditing Electronic Medical Records (EMR) exchange [2], real estate market to track ownership [3], etc. Despite such a variety of applications, the blockchain actually performs mostly the same tasks everywhere, since it is used for the same core functionalities across domains: *identity management* and *asset tracking*. In cryptocurrency applications, for instance, the asset is money; in supply chain, it is goods; in healthcare, it could be the EMR of patients. In all cases, the blockchain provides a way to track ownership of the asset while protecting integrity of owners' identity.

Regardless of the application scenario, most blockchain technologies (BCT) are becoming more flexible and expressive thanks to the notion of *smart contract* (SC): introduced by the Ethereum

platform [4] following the vision of Szabo [5], it is now available in some form in almost every other BCT. SC are user programs that can inspect and act on the blockchain as well as own digital assets, whose execution is performed, validated, and constrained by the underlying blockchain [6]. This is useful in several domains to let the blockchain deliver a third functionality: automatically *mediate* and regulate the interaction amongst off-chain entities—that is, human users or front-end software systems using a BCT as a back-end. In such a context, SC are written, deployed, and invoked by users to introduce a trusted mediator between two or more mutually untrusted parties willing to interact—in other words, to *coordinate* users' interaction. In fact, SC are considered as *trustworthy* given that their source code—therefore their operational behaviour—is publicly inspectable, and tamper-proof since it is stored on the blockchain.

A great deal of the literature about smart contracts is concerned with assessment of SC computational models and their *computational expressive power*—for instance by establishing whether a specific SC is Turing-equivalent or it guarantees termination. Yet, to the best of our knowledge, no paper assesses SC expressiveness on the *interaction dimension* [7] along which SC mediation operates. In other words, there has been not much attention on assessing the expressive power of SC and BCT from a *coordination perspective* [8], to determine for instance which system properties can be ensured by SC mediation.

Along this line, in this paper we aim at filling this gap by adopting a common approach in the field of *coordination models and languages* [9], that is, the premiere source of literature for issues related to interaction: encoding a model for which expressiveness is unknown (e.g., a specific SC) into one with known expressiveness (e.g., a well-known coordination model), so as to assess to which extent the latter (known) properties and expressive power is preserved by the former. As the target model to exploit as expressiveness benchmark our choice falls on LINDA [10], possibly the best-known coordination model in the literature, whose expressiveness and properties have been studied and assessed by many different works under many complementary perspectives [11–14].

Accordingly, the contribution of this paper is:

- to implement the LINDA coordination model as a SC on top of different BCT, so as to assess each SC expressive power along the interaction dimension
- to compare the different implementations in an attempt to sieve the fundamental mechanisms endowing a given BCT with the capability of successfully supporting LINDA properties and expressive power
- to propose a novel computational model for SC, implemented on a custom blockchain called Tenderfone (https://gitlab.com/pika-lab/blockchain/tenderfone/tenderfone-sc) [15], which outperforms the expressiveness of state of art SC along the interaction dimension

The remainder of the paper is hence structured as follows: Section 2 introduces the essential notions related to coordination models and languages, and overviews the main features of a coordination model in terms of requirements, design choices, and provided properties, taking the archetypal LINDA model as a reference; Section 3 describes the blockchain technologies considered in this study specifically regarding their abstractions and mechanisms lending themselves to a coordination-based interpretation, and focussing in particular on Tenderfone; Section 4 presents our implementation of LINDA on top of them, as a benchmark to assess their expressive power along the interaction dimension; Section 5 elaborates on the strengths and shortcomings of each implementation and, in particular, discusses why Tenderfone is better suited than other BCT to deal with coordination issues; Section 6 positions our contribution in the literature; finally, Section 7 concludes the paper.

## 2. The Coordination Perspective

The term "coordination" has an intuitive meaning in our everyday language, essentially defining the process of somehow handling activities relying on one another so as to make them all progress satisfactorily. As coordination is a term used in many different contexts, as many definitions can be found in academic literature. We here report three, which we take as reference for the whole manuscript:

- The former one is by Malone and Crowston [16], and purposely avoids references to any specific research field: they broadly define coordination as *"managing dependencies between activities"*. This definition puts the notion of dependency at the centre, and well fits SC functionality of mediating interactions amongst blockchain users, as the very existence of an interaction implies existence of a dependency (e.g., a user activity depends on the state of the blockchain, which is checked by the SC).
- The second one is by Gelertner and Carriero [17], and instead is explicitly bound to the context of programming languages and models: *"The coordination model is the glue that binds separate activities into an ensemble"*. This definition hints at existence of a space to be filled by something (the glue) in order to give cohesion, a shared purpose, an individual identity to originally separate (computational) activities. This is possibly difficult to see in a blockchain setting, yet many applications exploiting the blockchain actually do that: just think of a supply chain handled through the blockchain, where smart contracts take care of checking goods along transfer of ownership, and trigger appropriate actions on the appropriate entities (the user companies) when due—e.g., payment or sanctions depending on whether shipment of goods satisfies the terms encoded in a SC. In that context, SC actually binds separate activities (production, testing, packaging, shipment, assemble, payment, replacement, reimbursement, etc.) into an ensemble (the supply chain "workflow" involving all the different actors).
- The latter one is by Ciancarini [9], and once again refers to a precise context, that of software engineering: *"a coordination model provides a framework in which the interaction of active and independent entities called agents can be expressed"*. This definition focus on the representation of interactions, and on independency of participating entities, which both suit well the practice of using smart contracts to mediate interactions between mutually-untrusted parties: the actors interacting through the blockchain are most often completely independent organisations or companies, which agree on exploiting the blockchain exactly to represent the conditions for their interactions, the terms under which interaction is considered satisfactory, and the actions to trigger in case of compliance or not—actually akin to real-world contracts, to some extent.

Building upon these definitions above, we derive a new one so as to better highlight how looking at coordination models and languages may help check SC expressiveness and, possibly, improve it beyond the state of the art. We thus refine the definition of coordination as a two-steps activity with a specific goal: coordination amounts to *(i)* defining the dependencies amongst activities, and *(ii)* how to handle them, so as to avoid unwanted interference or undesirable side-effects, while achieving target properties at the level of the global coordinated system. As a result, we preserve the focus on representation of dependencies while also putting emphasis on the properties we want to endow the system with.

Many coordination models, languages, and technologies have been proposed for about thirty years now, especially in the multi-agent systems community—see [18,19] for a survey. Among those, *tuple-based coordination* [20] has been extensively studied mainly for its greater expressiveness and openness despite minimality w.r.t. approaches based on message-passing. Linda [10], in particular, is the archetypal tuple-based coordination model, and as such has been studied and extended throughout the years along many dimensions: formal expressiveness, suitability to diverse application domains, strengths and weaknesses, implementation techniques, etc.

The main elements of Linda are tuples, templates, tuple spaces, and communication primitives.

- A *tuple* is a data chunk represented according to a well-defined tuple language, specifying the structure of admissible tuples.
- A *template* is a concise way of representing a set of tuples: it consists of a pattern, represented according to a particular template language, which may be matched by several tuples. In other words, it is a blueprint for tuples, specifying possibly partially their content and structure.
- A *tuple space* is the repository where tuples may be put, inspected, and withdrawn by agents of any sort (e.g., processes, threads, software agents, human users interacting through an interface program).

- A *communication primitive* is an operation provided to such agents to put, inspect, and withdraw tuples from a tuple space. LINDA provides three primitives, whose semantics is depicted in Figure 1: `out` to put a tuple in a tuple space, `rd` to inspect a tuple in a tuple space, if any one matching a given template is present, and `in` to withdraw a tuple from a tuple space, if any one matching a given template is present.

What makes such a model suitable as a coordination language is the *suspensive semantics* of the aforementioned primitives: when `rd` or `in` do not find a matching tuple when they are issued, they suspend waiting for one. This simple mechanism alone straightforwardly enables processes to synchronize their actions based on availability of tuples.

For example, recalling the supply chain scenario mentioned above, it is easy to imagine adopting the LINDA model to ensure that every actor performs the necessary actions when due: items traversing the supply chain would be the tuples, each stage of the "supply chain workflow" would withdraw them from the tuple space, to actually inhibit progress of subsequent stages until the precedent ones have successfully completed, and actually enabling only the subsequent stages for which the needed items are available, and only when they become available. For instance, shipment would be active only after production, testing, and packaging, and only for those items actually packaged.

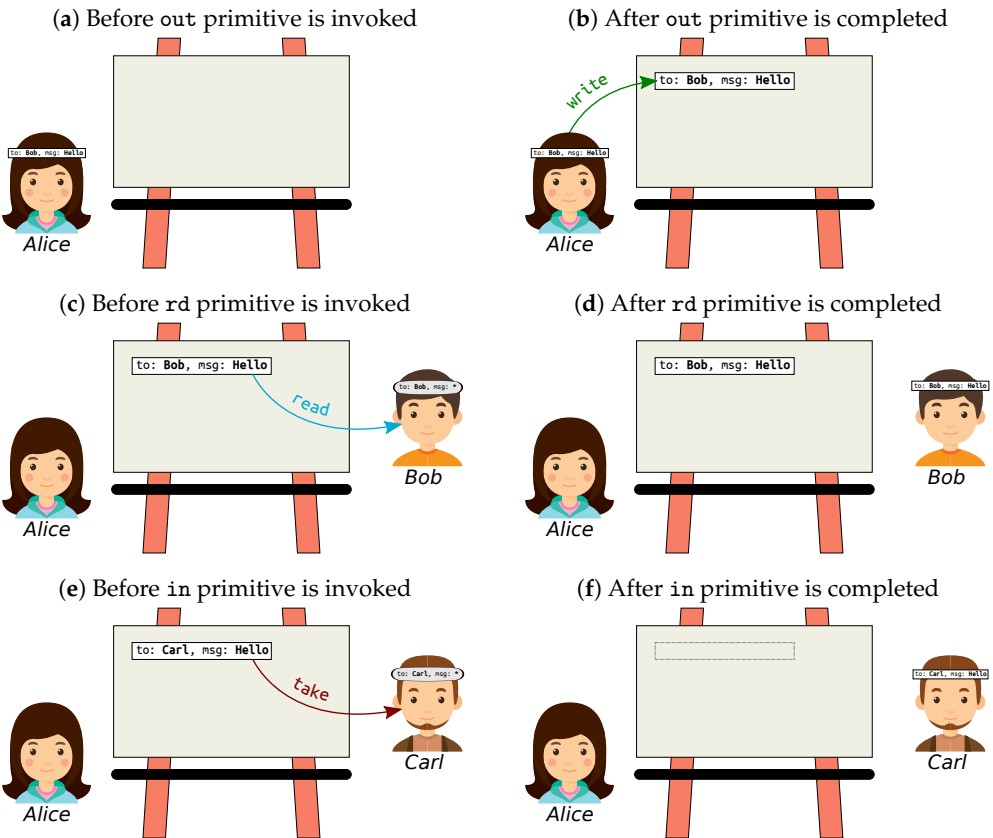

**Figure 1.** Graphical representation of LINDA primitives behaviour: tuple spaces are "blackboards" available to agents, which put (`out`, subfigure **a,b**), read (`rd`, subfigure **c,d**), and withdraw (`in`, subfigure **e,f**) data chunks called tuples.

Suspensive semantics is not the only relevant feature that characterises the LINDA model. For the purpose of this paper, enumerating the others serves the purpose of highlighting the requirements that they pose over LINDA implementations, the design choices they leave to implementors, and the properties they endow to the coordinated system, so as to better ground the discussion presented in Section 5 and make it possible to focus on the most impactful features of the implementations

described in Section 4. In particular, LINDA has two other distinctive features that define the model, and translate onto requirements for implementors.

**Generative communication**—tuples put in a tuple space have their own lifecycle, *independent* of that of the agents that produced them. In other words, once a tuple has been put in a tuple space, that tuple is equally accessible to any other agent, currently present or joining the tuple space in the future, until it gets explicitly withdrawn by an `in` primitive—by any agent, not necessarily the one who produced it. This means that to faithfully implement a tuple space, information must be stored on a dedicated repository whose lifetime can extend beyond that of interacting actors.

**Associative access**—tuples in a tuple space can be retrieved according to their content and structure through the template mechanism. In other words, in order to get access to specific tuples, agents need not to know the tuples name, location, address, or anything similar (in fact these notions do not even exist in the context of a tuple), but should only be able to specify, possibly partially, their content and structure by using suitable templates. This implies that a language for describing the content of information items is needed (a *template*), and that search and retrieval should be based on such a language.

Altogether, suspensive semantics, generative communication, and associative access enable on-demand, *data-driven synchronisation* of asynchronous computational activities of any sort. Furthermore, such a synchronisation mechanism—which straightforwardly enables arbitrary complex coordination protocols and patterns to be implemented—also provides for *reference*, *space*, and *time uncoupling*, as the interacting processes need not to know each other names, addresses, or references, nor to be co-located in space or time to successfully coordinate. This directly makes LINDA suitable for any open system in which participants may join/leave dynamically, and for any system demanding fault-tolerance—as faulty processes may simply try to coordinate again when they restart, as the tuple space has an independent lifecycle. All of this naturally fits the SC intended purpose of mediating interactions amongst mutually-untrusted parties: actors relying on the blockchain for this kind of coordination may simply care about the information they need to progress their own activities and comply to a protocol agreed with the other organisations joining the blockchain, while the blockchain itself (through a "LINDA SC") takes care of ordering actions depending on their reciprocal dependencies and available information stored on the blockchain (or ad-hoc metadata stored in SC). In our example above, in fact, the different companies along the supply chain simply care each about the items they need to perform the task they are responsible for (e.g., testing, packaging, shipment), and delegate the duty of guaranteeing that items progress along the supply chain only when due, hence tasks are triggered only when meaningful (e.g., shipment after testing), to the blockchain itself—or better, to the SC working as coordination medium.

The aforementioned requirements could actually be satisfied in different ways, hence a few design choices are available to LINDA implementors, for instance regarding the tuple and template languages—that is, how to describe data chunks and the blueprints to seek for them, the matching mechanism enabling associative access, as well as the actual data structures representing tuples and tuple spaces. Based on whether such design choices fully respect the requirements above, and how, the LINDA implementation at hand might either fully support or not the specific benefits of LINDA-based coordination described above—namely, *synchronisation* and *uncoupling*. Hence, we adopt satisfaction of such requirements as the metrics to measure expressiveness along the dimension of interaction of each specific blockchain. Thus, if a given BCT supports all the requirements and all the LINDA properties, it has maximal expressiveness, otherwise expressiveness decreases and the BCT is strictly less expressive than another one—that is, supports a stricly-smaller number of coordination protocols.

In Section 4 we discuss how the blockchain relates to the aforementioned requirements and design choices, hence whether a coordination model like LINDA can be fully modelled and implemented on top of currently available blockchain technologies. Before that, next section overviews the BCT we checked against the LINDA benchmark for assessing expressiveness of SC along the interaction dimension.

## 3. Selected Blockchain Technologies and Smart Contracts

Regardless of specific implementations, blockchain technologies (BCT) consist of a peer-to-peer network of nodes enacting a *consensus protocol* that lets them globally behave as a single *state machine*, aimed at consistently tracking modifications to data jointly manipulated by a number of clients. Handling those modifications called transactions (that is, the pre-conditions for their acceptance and the effects they bring about to the blockchain), as well as their *chronological ordering*, is the responsibility of special nodes of the blockchain, which commit the results of the operations to blocks of data linked by hash chains, following the principles described in [21]. This process creates a hard-to-tamper sequence of blocks—indeed, the *block*-chain—tracking the whole history of system evolution. Indeed, the interest around the blockchain lies essentially in this capability of maintaining a consistent *shared state* between mutually-untrusted parties.

The aforementioned state machine is not limited to a fixed program, but may as well be a general-purpose interpreter executing some custom program defined by external agents, as in the case of blockchain supporting the smart contract (SC) abstraction, which is indeed the program to be executed consistently by every node of the blockchain—which ultimately makes it behave as a general-purpose distributed computing platform. There, distributed consensus is exploited to ensure that all SC are consistently executed in the exact same way on all replicated nodes, provided they get the same input—namely, the same set of ordered transactions.

The many BCT currently available implement the concept of SC in many different ways. For instance, some of them are strict in the way new agents are admitted to be part of the system, thus allowing for finer access control mechanisms: they are therefore called "permission*ed*". Others let agents create their own identities, instead, usually by means of public keys, thus virtually letting anyone join the system: they are therefore called "permission*less*". From a strictly computational perspective, BCT also differ in *(i)* how they represent the shared replicated state, there including what sorts of information SC can store and handle; *(ii)* the admissible operations on such a state, there including rules concerning how SC can be instantiated, destroyed, or updated; *(iii)* which sorts of computation SC are able to perform, and on which data; *(iv)* how data is expected to flow from agents to SC—or among them—and vice versa. The remainder of this section describes the four specific BCT taken as a reference to assess SC expressiveness along the interaction dimension, and clarify the aforementioned aspects.

### 3.1. Ethereum

Ethereum [4] is a permissionless BCT with a native currency, the Ether (ETH), exploited in a Proof-of-Work (PoW) based approach to consensus, which is enacted by "miners"—i.e., the special nodes in charge of producing blocks.

Ethereum shared state consists of a number of *accounts*, each one associating information to either off-chain agents or SC. Such information include a *balance*, in terms of ETH—which implies that both agents and SC can own money—and, in case of SC, a *storage* area for anything that can be represented as byte strings. SC encapsulate custom and stateful *behaviours*, whose state consists of the aforementioned storage, whereas behaviour is expressed in terms of an ad-hoc bytecode which is stored on the blockchain as well—even if it cannot be modified. State may change as a consequence of *transactions* (TX) being published by off-chain agents. In particular, the admissible operations that transaction may convey are: *(i) deployment* TX to create new SC, *(ii) money transfer* TX to exchange money, *(iii) invocation* TX to send a message to a SC, triggering one of its behavior—whose effects may depend on both the message content and the current SC state. Once triggered, SC can interact with each other through *synchronous* function calls, and with off-chain agents through a publish–subscribe mechanism where the blockchain itself is used as a blackboard where events of interest are published by SC and read by off-chain agents.

Ethereum discourages long, possibly non-terminating computations in SC through the notion of *gas*, that is, a sort of fuel which must be included into TX by the agents publishing them. Gas is literally

bought by agents by consuming their ETH, and TX processing always consumes some gas, depending on which and how many bytecode instructions are actually executed. If the provided gas for a TX is not enough for it to complete, then its effects on the system state are completely reverted, except for gas consumption. Since the total amount of ETH is finite, there cannot be infinite computations, whereas long ones are still possible but made extremely costly—hence, discouraged.

As pointed out in [22] and briefly summarised in Section 4.1, this trait of Ethereum has consequences on expressiveness of Ethereum SC as far as the interaction dimension is concerned.

### 3.2. HyperLedger Fabric

HyperLedger Fabric [23] (HLF) is a permission*ed* BCT with no native currency, made of several peers (or nodes), possibly playing different roles, and the off-chain agents interacting with and through them.

One or more *ordering* nodes (or *orderers*) compose the *Ordering Service*, which is in charge of a consensus protocol of choice aimed at ordering the many transactions possibly issued by off-chain agents. Several blockchain data structures may be in principle created and stored by non-ordering peers in the form of *channels*, that share the view on the order of events given by the Ordering Service. A many-to-many relation binds channels and organizations.

A *Membership Service* is in charge of assigning peers to roles, verify their membership to a particular organisation, and define organisations' rights and roles w.r.t. channels. The set of organisations joining a channel, as well as the set of peers composing an organisation, are dynamic configurations of the HLF system, which may vary over time.

For a given channel, one or more *chaincodes* (i.e., smart contracts) may be dynamically installed, executed, or queried similarly to what happens in Ethereum. This is where custom business logic of organisations is injected. Chaincodes in HLF may be written in Go, JavaScript, or Java, and rely on the Shim API (https://hyperledger-fabric.readthedocs.io/en/latest/chaincode4ade.html). Similarly to what happens for Ethereum SC, the internal state of HLF chaincode is composed of key-value pairs containing arbitrary data. Such a state may evolve as a consequence of the chaincode being executed, as triggered by off-chain agents through requests (i.e., messages). Again, similarly to Ethereum, chaincodes may interact with each other through synchronous method calls, and with off-chain entities through a channel-wise event bus.

For each channel, one or more *endorsing* peers (or *endorsers*) must be defined—usually, at least one for each organisation—to execute chaincodes. More precisely, when a chaincode invocation TX is published, the corresponding code is *simulated* by as many endorsing peers as requested by that chaincode's *endorsement policy*—that is, essentially, a boolean formula stating *which* endorsers must approve the transaction. Only if the endorsement policy is satisfied—i.e., a sufficient amount of endorsements has been gathered—the transaction can be ordered by the Ordering Service and then, finally, registered on the channel. This last step is where the side effects possibly produced by the invocation TX are reified.

This distinction among endorsing and ordering nodes is a peculiarity of HLF, and defines its *execute-order-validate* architecture, described in [23], whose ultimate purpose is to prevent non-terminating chaincodes from starving the system. Roughly speaking, the advantage w.r.t. other BCT—which commonly adopt what the authors of [23] call the *order-execute* architecture—is that long-lasting or infinite computations performed by some chaincode would only starve the endorsing nodes of that chaincode, leaving the ordering nodes and the others nodes of the system unaffected.

### 3.3. Corda

Corda [24] is another permission*ed* BCT with no native currency, designed specifically for financial applications. A Corda system is made of several nodes, each identified by the system *doorman* service—i.e., a certification authority, which is the only entity capable of issuing *certificates* to cryptographically bind real-world organisations to some public key.

Corda is designed considering *privacy* as its very first concern, thus there is no such a thing like a global, shared state, by default: information, there including the asset tracked by the system and the transactions issued by nodes, is shared among nodes only if and when it is strictly necessary. Therefore, each node only perceives a portion of (or a point of view on) the system state, called *Vault*. This is possible thanks to the *Unspent Transaction Output* (UTXO) mechanism—which is strongly inspired to how BitCoin works [25], described below.

In Corda, the simplest chunk of information that can be represented is called *State*. States represent the assets that the Corda BCT keeps track of. The actual structure and content of States is application-specific, and must be defined by developers through the Java or Kotlin programming languages, before system deployment. The many nodes composing a Corda system may cause changes to be applied to each State, through transactions. By adhering to the UTXO mechanisms, transactions in Corda carry two sorts of information: *(i)* the set of States to be *invalidated* (a.k.a. consumed) by the transaction (i.e., its inputs), and *(ii)* the set of novel States to be created (i.e., its outputs). This mechanism iteratively builds a directed acyclic graph of transactions (depicted in Figure 2), whose unconsumed *fringe* represents the current state, and whose portion of interest for a given node represents its Vault.

SC in Corda are simply called *Contracts*: they are not dynamically deployed by users but by system administrators at deployment time.They are meant to check transactions validity w.r.t. their senders and their intended receivers, other than, of course, their input and output states. Even though such design choices may appear as constraining, they are actually quite flexible since States may represent virtually any data structure, and any sort of operations upon them, whereas Contracts simply define the contexts where such operations are admissible.

Any control-flow-related aspect involving the coordination of one or more nodes can be performed in Corda through *Flows*. A library of basic Flows to be composed is made available to the developers of Corda-based applications (https://docs.corda.net/api-flows.html). Flows are the place where articulated business logic is put, given the constraints affecting Contracts. For instance, as it is further discussed in section 4.3, we exploit Flows to mimic LINDA' suspensive semantics.

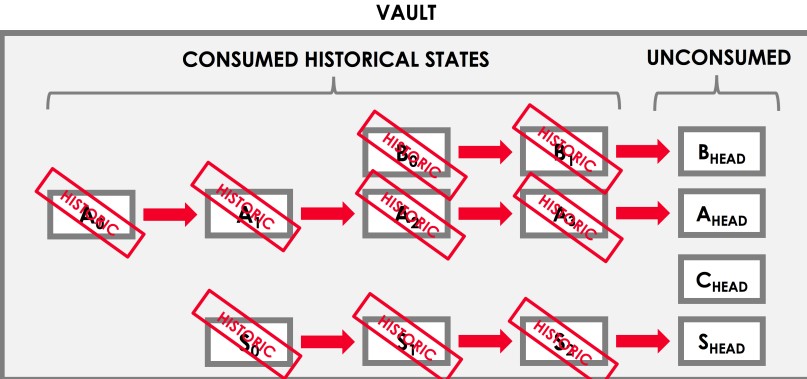

**Figure 2.** Graphical representation of the *Vault* (taken from https://docs.corda.net/key-concepts-states.html).

Transactions ordering and certification is the sole responsibility of *Notary Services*, very similar to the Ordering Service in HLF: a number of nodes enacting some consensus protocol. The main difference is that Corda allows multiple Notary Services within the same system, each possibly exploiting different consensus protocols. When a node issues a transaction, all the involved nodes must receive a copy of the transactions data to be signed. Then, the transactions must be notarized by one or more Notary Services, and finally it must be sent back to the involved nodes to let them update their Vaults.

### 3.4. Tenderfone

Tenderfone is the BCT introduced in [15] based on Tendermint (https://tendermint.com), a modular middleware supporting *state machine replication* for custom applications, through a byzantine fault tolerant consensus algorithm. Unlike other BCT, Tendermint is not equipped with any specific business logic: developers can use Tendermint to build their own BCT, by focussing only on which assets should be tracked on the blockchain, and how.

Being based on Tendermint, Tenderfone comes with no native cryptocurrency, and is permissioned. The core feature of Tenderfone is the notion of *pro-active smart contract*—as discussed in [26]. More precisely, SC in Tenderfone are endowed with a belief base and a plan base, both expressed through a declarative language based on First Order Logic (FOL)—in particular, Prolog. In particular, the SC belief and plan bases consist of *logic facts* and *predicates*, respectively. This is possible because all data in Tenderfone consist of *logic terms*, which can virtually represent arbitrarily complex data structures.

Smart contracts in Tenderfone compute through logic reasoning, aimed at *proving* some goals through deduction. Such goals can either be endowed into the SC upon creation or be provided by both off-chain users or other SC, through message passing. At a given point in time, the global, shared application state of Tenderfone consists of the set of all smart contracts' belief and plan bases, which is replicated consistently among system nodes by Tendermint. Such a state can be directly altered only by SC.

However, off-chain users can alter the system state indirectly, through a fixed set of operations, which allow them to *instantiate* new SC, *send* them messages, or *destroy* them—only a SC creator can destroy it. In order to update a SC state, a user sends a message to it, triggering some computation on that SC. Similarly, SC can alter the system state by interacting with each other, again by sending messages. In any case, the computations triggered by messages, on a SC, are the only means for updating its belief or plan bases—and of course SC retain a strict control on the admissible updates to their knowledge or behaviour. We thus say that in Tenderfone, SC *encapsulate* data, behavior, and control—hence they are conceptually modelled as agents.

It is worth highlighting the peculiarity of Tenderfone SC w.r.t. other BCT. In fact, they are able to interact with each other *asynchronously*, though message-passing. More precisely, when a SC, say `Alice` (in line with Figure 1), sends a message `M` to another entity, say `Bob`—which may be either an off-chain user or another SC, `Alice` does not want to wait for a response before progressing with its computation. From `Bob`'s perspective, this means that `Bob`'s computation triggered by `M` will start only *after* `Alice`'s computation—the one which produced `M`—is over. Of course, `Bob` is not required to answer immediately, or even at all. Such mechanism may be exploited by SC to delay, split, or postpone computations, and is the basis of the LINDA on Tenderfone implementation described in Section 4.4.

As far as termination is concerned, Tenderfone inherits Tendermint *order-execute* architecture, meaning that long-lasting or infinite computations may indeed starve the whole system. To prevent that from happening, Tenderfone interrupts and abruptly terminates all computations requiring more than a fixed amount of time—similarly to what happens in Cloud computing when temporal quotas are in place. However, SC in Tenderfone may still perform long-lasting computations by splitting them in steps and by attaching each step to a self-sent message, or by leveraging on time-related primitives introduced on purpose—as discussed in [26].

In next section, the four BCT described here are used to encode the LINDA model, so as to assess their expressive power along the interaction dimension. For each BCT, the features which helped or hindered the most implementation of LINDA are discussed.

## 4. Comparing BCT Expressiveness along the Interaction Dimension

In the following subsections we describe four proof-of-concept implementations of *blockchain-based coordination*—namely, we implement LINDA upon Ethereum (in two different ways), HLF, Corda, and Tenderfone. The aim is two-fold. On the one hand, we aim at assessing the expressiveness of each BCT w.r.t. dimension of interaction: this is unprecedented in the literature, in spite of the fact that SC are

commonly used to mediate interactions. On the other hand, we intend to devise out which features of each BCT are enablers or road-blockers for a notion of blockchain-based coordination, so as to focus there further works on the most promising BCT.

### 4.1. LINDA on Ethereum

Designing and implementing LINDA over the Ethereum platform has been the purpose of [22], hence we here summarize the relevant findings. There, a LINDA implementation is proposed where:

- tuple spaces are mapped to smart contracts
- tuples and templates are represented as raw strings
- the template matching mechanism coincides with string inclusion, for simplicity
- primitive *invocations* are performed through SC methods calls
- primitive *completions* are performed through Ethereum's publish–subscribe mechanisms

Also, the set of entities which can be coordinated coincides with the set of off-chain entities. In other words, coordinated entities are mapped to end-users only, not other SC—a dedicated section in Section 5 clarifies implications of such a limitation.

Ethereum SC fully support generative communication and associative access, as both the blockchain and SC internal storage enable tuples to have a lifetime independent from that of their producers. Suspensive semantics instead cannot be fully supported, as the Ethereum virtual machine *(i)* forbids infinite or arbitrarily long computations, thus, consequently, makes SC unable to wait for external events by polling for them iteratively, *(ii)* forces SC to interact with each other only through *synchronous* method calls, hence makes it impossible to separate the control flow across multiple activities so as to keep clients suspended while the SC itself processes further requests, and *(iii)* provides no facility for letting SC postpone their computations without external intervention of some off-chain agent, then cannot support suspending a computation to later resume it.

For the above reasons, the concept of SC currently supported by Ethereum cannot reach maximal expressive power along the interaction dimension, as it cannot implement the LINDA model by fully supporting its features. In fact, as a consequence of the above, synchronisation of activities cannot be fully supported, despite reference, space, and time uncoupling can be successfully preserved, instead.

An interesting peculiarity of Ethereum, stemming from its permissionless nature, is the fact that every coordination-related operation brings along an economic aspect to consider: who pays money for the operation. Since Ethereum is permissionless, it has a native cryptocurrency used to keep the blockchain working properly by incentivising good behaviours and discouraging malicious ones. This means that every operation on the blockchain costs money, which is then earned as compensation by miners. Coordination primitives are not excluded, and the most interesting aspect is that a specific implementation may impact the cost model adopted. In [22], for instance, we distinguish two models: an *advertisement model*, in which publishers of information (namely, who performs an `out`) pays more the more that the published information is sought for by other actors (which instead pay a constant amount to access information through `rd` or `in`); and a *market model*, in which publishing information has a constant price, yet accessing it costs more the more such information is unique (as in free markets scarce resources cost more).

### 4.2. LINDA on HyperLedger Fabric

The implementation of LINDA on HLF is similar to the one on top of Ethereum just discussed. Technical details are provided in the original conference paper [27], here we summarise the most relevant aspects so as to focus attention on the outcomes of the LINDA benchmark and on comparison with other implementations.

To encode the LINDA model on HLF:

- tuple spaces are mapped to *chaincodes*

- tuples and templates are represented as raw strings, whereas the template matching mechanism coincides with string equivalence—again, for simplicity
- primitive *invocations* are performed through chaincode invocations—that are, essentially, *synchronous* method calls
- primitive *completions* are attained through HLF Event Service (https://hyperledger-fabric.readthedocs.io/en/release-1.4/peer_event_services.html), that is, essentially, a channel-wise *publish–subscribe* mechanism
- coordinated entities are mapped to off-chain agents only

Despite the many architectural differences between HLF and Ethereum, the semantics of SC and chaincodes execution—as well as their API—are very similar in structure, purpose, and capabilities. Hence expressiveness along the interaction dimension is bound to be similar, too.

In fact: *(i)* even if chaincodes may *technically* perform infinite or long computations—thanks to the innovative *execute-order-validate* architecture—they still do not support busy-waiting on external events because of the transactional nature of chaincode execution; furthermore, *(ii)* similarly to the Ethereum case, chaincodes can *only* interact with each other through synchronous method calls, and *(iii)* they cannot suspend, schedule, or postpone their computations. Thus, HLF chaincode as well offers limited support to LINDA suspensive semantics, hence cannot reach maximal expressiveness as regards coordination.

It is worth mentioning that despite this similarity on assessment of expressiveness, the peculiar design and architecture of HLF provides for some interesting features, mostly involving access control and efficiency. For instance, by exploiting *endorsement policies*, it is possible to choose which and how many nodes are in charge of actually operating tuple spaces—namely, performing the computations associated with system functionalities. Also, the execute-order-validate architecture of HLF allows for actual parallelism when concurrent primitive invocations are being issued on the same tuple space—assuming that several endorsing nodes and an adequate endorsement policy are in place. The two features combined enable unmatched opportunities for scalability.

### 4.3. LINDA *on Corda*

The implementation of LINDA on Corda is instead quite a peculiar one, and considerably different from the two just discussed. Differences are mostly due to the particular features characterising Corda, such as Vaults and Flows. In our implementation:

- tuples coincides with States, which essentially wrap arbitrary strings
- templates are represented by strings as well
- the matching mechanism again coincides with string equality, for simplicity
- tuple spaces are not explicitly represented, as they coincide with Vaults, since Vaults consists of sets of unconsumed tuples, in turn
- primitive *invocations* and *completions*, again, are not explicitly represented, but they are obtained through Flows
- coordinated entities are represented by Corda's nodes

Mostly due to the peculiar notions of Vault and Flow, Corda has the least expressiveness along the interaction dimension amongst the three BCT considered. In fact, not only Corda shares the same limitation of Ethereum and HLF as regards LINDA suspensive semantics, but also *reference uncoupling* can no longer be supported: as Corda information is not shared among nodes by default, the nodes willing to share some States with other nodes must include the identifiers of such nodes into the transactions generating the shared States. This implies that in order to publish some information, a node should know the identifiers of all the other nodes that are expected to *perceive* that such an information exists in the first place.

### 4.4. LINDA *on Tenderfone*

Being an unpublished contribution, the source code of LINDA on Tenderfone is fully shown in Listing 1. The implementation is structurally similar to the HLF and Ethereum based ones, even if its semantics, behavior, and capabilities are different. In fact, it is characterized by the following conceptual mapping:

- tuple spaces are mapped to smart contracts
- tuples and templates are represented as FOL terms,
- template matching mechanism coincides with *logic unification*—which is more expressive than the other mechanisms exploited so far
- both primitive *invocations* and *completions* are mapped to *asynchronous messages*—which in Tenderfone can be sent by both off-chain agents and SC
- the set of coordinated entities comprehends both off-chain agents and SC

Here, the novelties w.r.t. other BCT lie in the way that Tenderfone implementation realizes LINDA suspensive semantics, other than the wider range of entities which can use it.

As one can understand by inspecting the Prolog code in Listing 1, the proposed solution states that a Tenderfone-based tuple space consists of a SC which listens for messages in the forms `rd(Template)`, `in(Template)`, or `out(Tuple)`—where `Template` and `Tuple` are logic variables which may match any sort of logic term. Apart from the internals of the proposed implementation—which are essentially aimed at properly handling stored tuples and pending primitives invocations—what is important to understand is the implications of switching to asynchronous message-passing among SC. In fact, the many lines in the form `send(Something, Agent)`—which represent the completion of some LINDA primitive—are *not* method calls, but messages that depart from the tuple space only *after* the `receive`/2 procedure containing them is over. This mechanism is what makes a symmetric design of tuple spaces possible: both primitive invocations and completions are modelled as messages. Also, in Tenderfone both off-chain users and SC may be the receivers of the aforementioned `send(Something, Agent)` messages representing primitive completions. This is not possible in the other BCT analyzed in this paper, where there is no way to implement a notion of primitive completion that would at the same time be *(i)* compatible with LINDA suspensive semantics, and *(ii)* accessible to smart contracts.

This feature of Tenderfone is indeed very powerful, as it is the foremost responsible for its greater expressiveness reach w.r.t. the other BCT described above: Tenderfone in fact has the greater expressiveness along the dimension of interaction, as it allows for a faithful encoding of the LINDA benchmark supporting all its properties. For example, a smart contract may be exploited to mediate and reify the interaction among other (off-chain or on-chain) entities. This enables the creation of articulated interaction, synchronization, or coordination patterns, such as the *barrier* (https://en.wikipedia.org/wiki/Barrier_(computer_science)) shown in in Figure 3. There, a Tenderfone SC acts as a LINDA tuple space, on top of which another one ($SC_2$) implements a *barrier* waiting for two events to occur before performing some activity. The two events consist of any two other entities—another SC and a user in this case—performing an `out(event(N))` operation, where $N \in \{1, 2\}$. The added expressiveness of Tenderfone lies in the fact that while $SC_2$ waits for its activation conditions, any other entity (that is both users and other SC) may perform other activities.

**Listing 1.** A Tenderfone SC acting as a LINDA tuple space. Predicates such as `findall`/3, `member`/2, and `delete`/3 are standard Prolog predicates, whereas predicates `send`/2, `set_data`/2, and `get_data`/2 are Tendefone built-ins described in [15].

```prolog
init(_) :-                                                          1
    set_data(tuples, []),                                           2
    set_data(pending, []).                                          3
                                                                    4
receive(rd(Template), Agent) :-                                     5
    find(tuples, Template), !,                                      6
    send(Template, Agent).                                          7
receive(rd(Template), Agent) :-                                     8
    add(pending, request(rd, Template, Agent)).                     9
                                                                    10
receive(in(Template), Agent) :-                                     11
    remove(tuples, Template), !,                                    12
    send(Template, Agent).                                          13
receive(in(Template), Agent) :-                                     14
    add(pending, request(in, Template, Agent)).                     15
                                                                    16
receive(out(Tuple), Agent) :-                                       17
    send(Tuple, Agent),                                             18
    findall(Request, find(Name, Request), AllPending),              19
    handle_pending(Tuple, AllPending).                              20
                                                                    21
handle_pending(_, []).                                              22
handle_pending(Tuple, [request(in, Tuple, Agent) | _]) :-           23
    send(Tuple, Agent).                                             24
handle_pending(Tuple, [request(rd, Tuple, Agent) | Others]) :-      25
    send(Tuple, Agent),                                             26
    handle_pending(Tuple, Others).                                  27
                                                                    28
%% Library predicates                                               29
find(Name, Item) :-                                                 30
    get_data(Name, List),                                           31
    member(Item, List).                                             32
                                                                    33
add(Name, Item) :-                                                  34
    get_data(Name, List),                                           35
    set_data(Name, [Item | List]).                                  36
                                                                    37
remove(Name, Item) :-                                               38
    get_data(Name, List),                                           39
    delete(Item, List, NewList),                                    40
    set_data(Name, NewList).                                        41
```

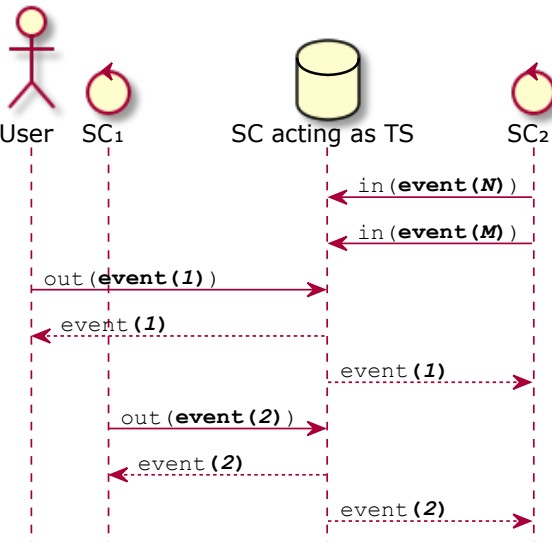

**Figure 3.** Tenderfone smart contract implementing a LINDA tuple space exploited to coordinate two other smart contracts according to the barrier pattern. Straight lines represent primitive invocations, whereas dashed lines represent primitive completions.

## 5. Discussion

Out of the four BCT checked against the LINDA benchmark, only Tenderfone has shown maximal expressiveness along the dimension of interaction. HLF and Ethereum are right behind, as they do not fully support LINDA suspensive semantics, despite all other properties are preserved. Corda falls behind with the least expressiveness as also reference uncoupling is compromised by Corda mechanisms to guarantee privacy.

The core mechanisms that differentiate Tenderfone from the state-of-art are *(a)* encapsulation of the control flow by SC, *(b)* asynchronous communication, and *(c)* pro-activeness [26]. Indeed, the comparison performed in Section 4 points out how currently-available BCT cannot satisfy such requirements. We argue that such limitation is mostly due to the poor choice of Object-Oriented Programming (OOP) as the computational model of smart contracts, which is often combined with the strict requirements on programs termination virtually affecting all BCT. In fact, it is widely recognised within the multi-agent systems community how OOP is not the best suited programming paradigm for interaction-related use cases, as objects do no encapsulate control flow, and are constrained to interact through synchronous request-response mechanisms.

Conversely, Tenderfone adopts a different programming paradigm—namely, the *agent-oriented* one—where SC can send messages to each other in an *asynchronous* way, with no need to wait for a response. Furthermore, the Tenderfone notion of proactive SC opens to the possibility of letting SC be triggered by other SC whenever a message is received, and with time triggers. This is different from mainstream BCT, where SC are strictly reactive to stimuli coming exclusively from off-chain entities.

When it comes to coordination issues, such mechanisms prove themselves to be more than mere technicalities. In fact, as demonstrated by our LINDA benchmark, asynchronous message-passing fully supports the construction of an expressive coordination model such as LINDA, which in turn widens the spectrum of real-world activities that can potentially be automated through SC.

Besides these results concerning expressiveness alongside the interaction dimension, hence supporting the SC role of mediating interactions, in the remainder of this section we aim at pointing out a few remarks which may help clarify why such an investigation is generally relevant.

### 5.1. On On-Chain Coordination

Let us consider the case of a smart contract *R* aimed at managing the many, concurrent accesses to the shared resources owned by some company—like, for instance, cars. In particular, let us suppose that

*R* is in charge of tracking and constraining the reservation and the release of resources performed by employees. This means employees can freely request the reservation of some resource for themselves, in a mutually exclusive way, by sending messages to *R*. When satisfying a reservation request, *R* takes care of registering the employee's identity as well, before answering to its message. Of course, employees are assumed to eventually release any reserved resource of theirs.

Let us suppose now that some employee is willing to automate the reservation procedure for her favourite car, in order to increase the chances of using that particular car. This could be achieved by deploying some software component *A* which is in charge of performing the reservation as soon as possible. There are many possible solutions to this problem, and each one may possibly leverage on different architectures.

For example, *A* may be an off-chain program working on behalf of a human being. Or, *A* may be yet another smart contract, as SC are programmable and user-defined. In the former case, the internal behaviour of the program would not be tracked by the BCT, hence should be trusted a-priori to comply to the terms and conditions of resource usage. In the latter case, instead, the internal functioning of *A* would be tracked by the underlying BCT used by *R*, thus making *A* amenable of public scrutiny—and, therefore, trustworthy. This leads to the notion of *accountable interaction* discussed in next section.

There would be also other considerable advantages in letting *R* behave specifically like a LINDA tuple space where tuples represent currently available resources. There, users would be able to reserve resources through the `in` primitive, or release them, through the `out` primitive. Of course, users could only consider a resource as reserved after the corresponding `in` primitive completes its execution. For example, this would enable *A* to reserve a car by means of the `in` primitive. In fact, thanks to LINDA's suspensive semantics, *A* could invoke the `in` primitive even if no car is currently available, knowing that the primitive will complete as soon as the requested car becomes available.

The fact of relying on a well-studied and formalised coordination model like LINDA would enable SC to be scrutinized in a rigorous way for properties to enforce and guarantee, and developers may have a more precise idea of what is possible or not with a specific BCT, with no need to start tinkering with the technology.

*5.2. On Accountable Interactions*

We now intend to elaborate on the benefits of using a coordination model such as LINDA as a general-purpose coordination service in any blockchain, hence going beyond its utilisation as a benchmark to measure expressiveness.

In current practice of using smart contracts as mediators of interaction limited to off-chain actors, the granularity of what can be actually tracked on the blockchain is limited to interactions of off-chain entities, and invocations of smart contracts, in general. Everything that happens as part of the computational cycle of a smart contract which does not generate any invocation, cannot be registered on the blockchain—as it generates no transaction. This means that each smart contracts developer decides the granularity at which coordination operations are inspectable and recorded.

Instead, by adopting a general-purpose coordination model such as LINDA to realise any other coordination protocol or policy, the granularity at which interactions are tracked by the blockchain is fixed ones and for all, and is the finest possible, as LINDA operations are the minimal set which guarantees maximal expressiveness along the interaction dimension. This brings us to a notion of *accountable interaction* which well suits the blockchain purpose. As the blockchain is exploited to track manipulation of a specific asset, with the goal of making actors accountable for their actions (amongst the many), likewise the blockchain may be exploited to track coordination amongst smart contracts and off-chain entities, making both accountable for their interactions.

## 6. Related Works

Research on blockchain and smart contracts is as diverse and abundant as the many facets of these technologies: for instance, programming model of smart contracts, opportunities for their concurrent

execution, application of blockchain to heterogeneous business domains, and novel blockchain models featuring ledger sharding, network partitioning, and other innovations are all subjects of active research. However, when restricting the research domain to the focus of this paper (coordination as enabling and constraining interaction [8]), literature is no longer so rich: the term "coordination" is often used with a much broader acceptation that the (stricter, and arguably more proper) one here adopted, since it often encompasses mere communication, data exchange, and simple sequential execution pipelines. Nevertheless, there are some works trying to implement some forms of coordination upon the mechanisms provided by the blockchain and smart contracts, which can be roughly positioned against our own work.

In [28], for instance, the authors recognise the fact that smart contracts can be used by entities to engage in contractual commitments, hence lend themselves (and the blockchain) to be used as coordination infrastructure. Differently from the perspective adopted here, they adopt an *institutional perspective* over coordination, thus rely on the Grammar of Institutions [29] as a language to express coordination laws, which revolves around 5 statements abbreviated as ADICO: *(A)* Attributes describe an entity characteristics; *(D)* Deontic describes the nature of the statement (obligation, permission, or prohibition); *(I)* Aim describes the action or outcome regulated; *(C)* Conditions describe the preconditions for the statement to hold; and *(O)* Or else describes consequences for non-compliance. Based on that grammar, the authors propose a mapping from ADICO to Solidity (the reference language for Ethereum smart contracts) constructs, along with a technique for automatically translating "programs" written in ADICO to smart contracts written in Solidity. Their work is an interesting first step towards re-interpreting smart contracts and blockchains as virtual institutions regulating interaction among off-chain entities (thus, coordinating them): nevertheless, no actual coordination-related use case is presented, as the voting scenario described is just about counting and triggering sequential activities rather than supporting complex form of coordination (as LINDA does, instead).

The work in [30] revolves instead around an actual use case and scenario—that is, using Ethereum smart contracts to implement *auction-based coordination* in the energy market scenario. In particular, a SC is defined to enable entities producing (excess) energy to sell it to the highest bidder (energy demand) through a Vickrey auction—which is known to encourage honest bidding. The core working logic of the proposed smart contract is as follows: when some energy producer wants to start an auction, it activates the contract and sets up an internal (off-chain) timer for bid acceptance. Then, buyers may bid through a dedicated function of the smart contract, which guarantees confidentiality of the bid until the auction (details are out of scope here, see the paper for a thorough description).

The authors of [31] propose to consider the blockchain as a *software connector* [32] alternative to existing centralised shared state storage means, thanks to its inherent decentralisation. Such a stance is akin to ours as connectors are defined as an architectural component enabling and constraining interaction, such as pipes, repositories, and sockets, with the goal of equipping distributed systems with desirable properties such as performance, reliability, security. Despite the idea is appealing, the paper does not described any specific definition of coordination based on blockchain: simply, smart contracts are re-interpreted as a software connector delivering coordination services, but actual mechanisms are not detailed, nor an example of smart contract reifying the idea is delivered.

In [33] yet another perspective over blockchain-based coordination is adopted, stemming from the research area on business process management: the proposition is that of adopting Ethereum smart contracts to implement *BPMN choreographies*, therefore using the blockchain as a collaborative processes facilitator. The core of the idea relies on a smart contract representing a choreography template and a pool of other smart contracts representing actual activities: upon creation, the initial activities in the choreography are enabled, then, when a collaborator calls the method corresponding to an enabled activity, the transaction is verified and if successful the choreography state is updated—i.e., the executed activity is disabled and subsequent ones are enabled. The authors propose a two-steps process to automate translation of a business process into Solidity code: first, the BPMN model is translated to a Petri Net according to the transformation method described in [34], then the Petri Net is translated to

Solidity code: namely, the resulting SC uses *(a)* two variables stored on the blockchain to encode the current marking of the Petri Net and to encode the predicates attached to transitions therein, *(b)* public functions corresponding to user tasks, and *(c)* the step function which fires all enabled transitions until it gets to a point where either a new set of user tasks are enabled or the instance has been completed. Although no specific example is given, the encoding of a BPMN model to a SC proposed by the authors seems to directly enable workflow-like coordination on the blockchain. Nevertheless, it is not clear how they address the issue of users (either off-chain software or humans) suspending to wait on completion of tasks: should they attempt to act and receive a failure if the task is not yet enabled according to the current state of the choreography, or have they some means of either being notified when or suspend until the task becomes active? These kinds of details make a great difference when attempting to use the blockchain as an effective coordination means, and are typically far from trivial to achieve (as we demonstrated by our LINDA-based benchmarks).

One last related contribution deals with *multi-agent planning* in cyber-physical systems. In [35] the authors propose to exploit the blockchain as a means to coordinate execution of shared plans between a set of agents, each in charge of specific activities which may depend on others. The approach described to achieve that is similar to that of [33]: a template smart contract represents the shared plan, and a pool of "satellite" smart contracts represent activities composing the plan. In particular, the authors rely on a smart contract (Plan_SC) which is the "scheduler agent" that orchestrates the plan, while a bunch of smart contracts (Act_SC) are each responsible for either sensing data from sensor agents/devices or dispatching commands to actuator agents/devices. Since these actions need to access an environment (the cyber-physical system itself) external to the blockchain, they are performed through an Oracle smart contract (Oracle_SC) exploiting the Provable Oracle service (formerly known as Oraclize) (https://provable.xyz). The core idea of the shared plan smart contract is as follows: the plan is represented as a list of lists, the element of each sub-list denoting an action; whenever an action is completed (by invocation of the corresponding smart contract), this is deleted from all the lists; thanks to the shared blockchain, the participant agents share a common list of actions, hence everybody is consistently updated about the state of the shared plan (which actions to do next, which are already completed, etc.). Seemingly then, blockchain-based coordination in execution of shared plans seems possible, although *(a)* no specific scenario is implemented and discussed, and *(b)* in the same way as in the previous related work [33] it is not clear how much polling on the blockchain the interacting agents have to perform in order to make the joint plan advance.

Summing up, despite the fact that a direct comparison with the Tenderfone-based approach is currently impossible—either for the lack of fully implemented benchmarks or for the radical difference of approaches—all the aforementioned works witness the interest in using the blockchain as the backbone of a coordination infrastructure, and motivates the need for further research along the line.

## 7. Conclusions and Further Work

In this paper we discussed implementation of the archetypal LINDA coordination model on top of four different blockchain technologies: three taken as the state-of-art (Ethereum, HyperLedger Fabric, and R3 Corda), one as our own contribution, so as to *(i)* assess the expressive power of current smart contract implementation along the interaction dimension, which is unprecedented in the literature, *(ii)* better understand which mechanisms hinder or satisfy the requirements of coordination models, and *(iii)* assess the potential of *blockchain-based coordination*.

Our results show that the variety of coordination protocols and models that can be faithfully implemented in any given blockchain is not to be taken as granted, as each blockchain exhibits a different expressive power alongside the interaction dimension—as demonstrated by our LINDA as a benchmark investigation.

Future works will be devoted at further exploring the issue, for instance by encoding state-of-the-art coordination protocols and patterns on top of our Tenderfone implementation of

LINDA, to then demonstrate their effectiveness in orchestrating real-world computations in any reference scenario—for instance, supply chain.

**Author Contributions:** Conceptualization, S.M., G.C., and A.O.; methodology, S.M. and G.C.; software, A.M. and G.C.; writing—original draft preparation, G.C., S.M., and A.M.; writing—review and editing, S.M. and A.O.; supervision, A.O. All authors have read and agreed to the published version of the manuscript.

**Funding:** This research received no external funding.

**Acknowledgments:** The authors would like to thank student Michael Bosello for authoring a related work [27] and for implementing a first prototype of the Fabric and Corda implementations of LINDA.

**Conflicts of Interest:** The authors declare no conflict of interest.

## Abbreviations

The following abbreviations are used in this manuscript:

SC　　　Smart Contract
EMR　　Electronic Medical Records
BCT　　Blockchain Technology
ETH　　Ether (the cryptocurrency adopted by Ethereum)
POW　　Proof of Work
TX　　　Transaction
HLF　　HyperLedger Fabric
UTXO　Unspent Transaction Output
FOL　　First Order Logic
w.r.t.　　Wit Respect To
OOP　　Object-Oriented Programming

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
