# Peer review of "Blockchain-Based Coordination: Assessing the Expressive Power of Smart Contracts†"

_information, doi:10.3390/info11010052_

Round 1

Reviewer 1 Report

This paper is about usage of blockchain technology for coordination of interactions between software agents. 

In the abstract, the authors claim that they "compare the coordination ... power of different smart contract models..."  However, I failed to find the results of this comparison.

What are the metrics? Section 4 (Comparing BCT...) contains some description of different technologies without any further comparison of these.

The discussion section is too short and too general, and does not address the paper's contribution. It is difficult to decide about the soundness of the work because the results are not clearly presented.

Further,  the Conclusion and Future work section does not really say anything about the future work. The Abbreviation section does not contain all abbreviations used in the paper. 

Author Response

First of all, we would like to thank the referee for his insightful comments which helped us sensibly improving the quality of the manuscript.

Here follows a detailed reply to each of the referee's concerns.

In the abstract, the authors claim that they "compare the coordination ... power of different smart contract models..."  However, I failed to find the results of this comparison.

Results are now more clearly stated in each subsection of section 4, by explicitly stating the degree of expressiveness supported by the specific blockchain analysed.

What are the metrics? Section 4 (Comparing BCT...) contains some description of different technologies without any further comparison of these.

The metrics to measure expressiveness along the interaction dimension have now been explicitly identified as the Linda properties in section 2, and further referenced during the comparison in section 4.

The discussion section is too short and too general, and does not address the paper's contribution. It is difficult to decide about the soundness of the work because the results are not clearly presented.

The discussion section has been expanded to better summarise the obtained results and convey the contributions stated in the introduction, while still not overlapping with the contents of the original conference paper, which provided for more technical considerations.

Further,  the Conclusion and Future work section does not really say anything about the future work.

We added description of our ongoing efforts along this research line.

The Abbreviation section does not contain all abbreviations used in the paper. 

We carefully checked that all abbreviations are correctly reported.

Reviewer 2 Report

This paper compares the coordination expressive power of different smart-contract models by adopting the archetypal LINDA coordination model as a benchmark. Also, authors propose a blockchain implementation considering a new definition of smart contract overcoming the studied limitations.

Section 3 “Blockchain technologies and Smart Contracts” should be reduced and oriented to their ability to implement coordination models.

In section 4 (main section of the paper) the authors “describe four proof-of-concept implementations of LINDA” on Ethereum, HLF, Corda, and finally on Tenderfone. However, the first three implementations are already published in [10] and [20], thus, authors evaluate (discuss) in a qualitative way which are the requirements, direct mappings and main problems.

In the case of Linda on Tenderfone, authors provide a SC and analyze the implications. However, this contribution seems scarce to me. In my opinion the paper would be much more complete and would be more novel if a more analytical / quantitative comparative of the implementations were made.

Authors are encouraged to provide a study of the performance of these solutions in terms of cost, time or amount of information to be transmitted. Also, a real use case could be used as an example of a situation in which a tuple-based coordination model is useful for blockchain interactions. In this way authors can argue the benefits of having a system such as Linda running on top of blockchain.

Please provide a profound English revision and revise the following typos:

Line 50, LINDA is mentioned, but it is now explained. A proper introduction of LINDA must be done here.

Line 235. the the simplest

LINE 292: though -> through

Author Response

First of all, we would like to thank the referee for his/her insightful comments which helped in sensibly improving the quality of the manuscript.

Here follows description of the improvements we made to satisfy the referee's concerns.

Section 3 “Blockchain technologies and Smart Contracts” should be reduced and oriented to their ability to implement coordination models.

We tried to remove unnecessary sentences while still preserving self-containment of the manuscript.

In the case of Linda on Tenderfone, authors provide a SC and analyze the implications. However, this contribution seems scarce to me. In my opinion the paper would be much more complete and would be more novel if a more analytical / quantitative comparative of the implementations were made.

We better identified the metrics upon which the described implementations are compared against in section 2, which correspond to the main Linda features. Then, we explicitly commented on such metrics during each comparison in section 4, and summarised the results about the expressive reach of each implementation.

Authors are encouraged to provide a study of the performance of these solutions in terms of cost, time or amount of information to be transmitted.

We understand the point of the reviewer, but we believe that such a comparison would be against the declared goal of the paper, which is to compare expressiveness of smart contracts models along the dimension of interaction, not from a computational performance standpoint---which would be not entirely new in the literature. We take the blame of not being clear about such a goal, hence we improved the introduction accordingly.

Also, a real use case could be used as an example of a situation in which a tuple-based coordination model is useful for blockchain interactions. In this way authors can argue the benefits of having a system such as Linda running on top of blockchain.

Although, as for previous comment, the goal of the paper is not entirely to convince about usage of Linda in the blockchain, but to exploit Linda to assess expressiveness along the interaction dimension, we added considerations along the line mentioned by the referee in the discussion section, as well as in the introduction. Again, we take the blame for not making this clear enough since the very beginning, hence improved the introduction accordingly.

Round 2

Reviewer 2 Report

Currently I'm satisfied with the corrections and improvements made by the authors. My comments were correctly addressed and paper was accurately refocused towards assessing expressiveness.

I think the current version of the paper can be accepted.